# Research on a Framework for Chinese Argot Recognition and Interpretation by Integrating Improved MECT Models

**DOI:** 10.3390/e26040321

**Published:** 2024-04-06

**Authors:** Mingfeng Li, Xin Li, Mianning Hu, Deyu Yuan

**Affiliations:** 1School of Information and Network Security, People’s Public Security University of China, Beijing 102206, China; 2Key Laboratory of Security Prevention Technology and Risk Assessment of the Ministry of Public Security, Beijing 100038, China

**Keywords:** argot recognition and interpretation, information entropy, semantic space, MECT model, transformer architecture, large language model, prompt engineering, DBSCAN

## Abstract

In underground industries, practitioners frequently employ argots to communicate discreetly and evade surveillance by investigative agencies. Proposing an innovative approach using word vectors and large language models, we aim to decipher and understand the myriad of argots in these industries, providing crucial technical support for law enforcement to detect and combat illicit activities. Specifically, positional differences in semantic space distinguish argots, and pre-trained language models’ corpora are crucial for interpreting them. Expanding on these concepts, the article assesses the semantic coherence of word vectors in the semantic space based on the concept of information entropy. Simultaneously, we devised a labeled argot dataset, MNGG, and developed an argot recognition framework named CSRMECT, along with an argot interpretation framework called LLMResolve. These frameworks leverage the MECT model, the large language model, prompt engineering, and the DBSCAN clustering algorithm. Experimental results demonstrate that the CSRMECT framework outperforms the current optimal model by 10% in terms of the F1 value for argot recognition on the MNGG dataset, while the LLMResolve framework achieves a 4% higher accuracy in interpretation compared to the current optimal model.The related experiments undertaken also indicate a potential correlation between vector information entropy and model performance.

## 1. Introduction

The 51st Statistical Report on the Development of China’s Internet [1] shows that, in 2022, a total of 845 million netizens participated in the purchase and sale of compliant items on online platforms, accounting for 79.2% of the total number of netizens. However, including the dark web and underground forums, illicit transactions persist, posing significant challenges to cyber and social security stability in cyberspace [2]. According to statistics, in its 2.5 years online, the dark web Tor site “Silk Road” amassed 150,000 users and transactions totaling $1.2 billion [3]. The online trading products of the dark web often include illegal and irregular items, such as drugs, electronic fraud materials, hacking tools, and smuggled goods [4]. To avoid scrutiny, industry practitioners often conceal sensitive content within argots mixed with normal content, enhancing transaction concealment. Figure 1, illustrates some hidden words and their explanations in the field of drug trading. In the first example, “宵夜” (midnight snack) refers to “毒品” (drugs), and “猪肉” (pork) refers to “冰毒” (methamphetamine).

The research on argot recognition and interpretation in combat units has a long history. Early law enforcement agencies used manual construction of an argot knowledge base to interpret known argots. For example, the US Drug Enforcement Agency (EDA) intelligence department developed a set of drug codeword libraries to decipher the collected evidence and data containing codewords [5]. Ouyang et al. collected and summarized relevant drug code libraries based on the language characteristics and ethnic customs of the Guangxi region [6]. Ouyang et al. also used railway property infringement criminals as their target for obtaining secret language, and summarized a set of railway property infringement secret language libraries [7]. After summarizing a substantial number of hidden language samples, previous research findings, and the results of our preliminary experiments, we have identified the following characteristics of Chinese argots:Chinese argots and the words they refer to (referred to as pronouns) are normal vocabulary, rather than special characters similar to Morse codes [8].There are inherent connections between Chinese argots and their pronouns, including their shape, pronunciation, and meaning, which are relatively loose and often unknown to outsiders [9].Most pronouns are nouns or verbs, but the part of speech of Chinese argots and pronouns may not be the same, and verbs and adjectives are often used to refer to nouns or verbs.If the lexicon of argot words is concealed, and multiple alternative words are provided for that position, the concealment capability of argot words and the entropy of the set of alternative words for filling in that position are positively correlated.

As the aforementioned algorithm only recognizes a partial set of argot features, the various algorithms mentioned earlier are now inadequate for the current scenario of argot recognition and interpretation, particularly in the task of argot recognition. Therefore, this paper integrates the concept of semantic space, drawing inspiration from the manifold assumption in deep learning. Additionally, it combines the notion of vector information entropy to assess the rationality of word vectors within the semantic space [10].

Specifically, this paper makes the following contributions in the domain of argot recognition and interpretation:We constructed a Chinese long text corpus MNGG dataset using an open-source cant dataset [8] to support research on Chinese argots recognition.A Chinese argots recognition model CSRMECT was proposed based on the MNGG dataset, MECT4CNER, and DBSCAN clustering algorithmsBased on the MNGG dataset, the large language model, and prompt engineering, a Chinese argot interpretation framework LLMResolve was constructed to carry out Chinese argot interpretation work.We built a framework for Chinese argot detection, combined with CSRMECT and LLMResolve, to construct a comprehensive cold start Chinese argot recognition and interpretation workflow covering all fields.

## 2. Related Work

In the early stage of NLP research in academia, the field most related to Chinese argot recognition was called Chinese morphs decoding and resolving. This field focuses on researching the bypass mechanism of sensitive word detection algorithms. Specifically, in order to avoid detection by detection algorithms, users often replace a sensitive word with another word. The replaced word is generally called a reference, and the word used to replace the reference word is called a morph [11].

In the field of Chinese morph interpretation, Huang et al. [12] conducted groundbreaking research, first proposed the concept of morphs, and constructed a morph dataset through Weibo. They also designed various algorithms to interpret morphs. Zhang et al. [13] used Huang et al.’s variant definition algorithm for morph interpretation. Then Zhang et al. [14] constructed a deep neural network-based interpretation algorithm and first proposed the concept of resolve candidate words. Sha et al. [15] proposed a framework based on word embedding for morph resolution. You et al. [16] proposed a variant interpretation method based on an autoencoder combined with contextual information, and the model performance exceeded that for all the aforementioned indicators for morph interpretation. In the field of Chinese morph extraction, Zhang et al. [13] designed various morph generation algorithms by analyzing the construction logic of morphs. They attempted to use these algorithms to generate morphs and used SVM-based detection algorithms for morph extraction, achieving good detection bypass effects. Afterwards, Zhang et al. [14] proposed a morph recognition algorithm based on SVM- and graph-based semi-supervised learning approaches. The morph recognition algorithm achieved an F1 value of 83% on the Weibo dataset designed by Huang et al. [12].

However, there are significant differences between the fields of Chinese argot recognition and interpretation, as well as Chinese morph recognition and interpretation. From the perspective of the research subject, pronouns in the field of variant recognition only include sensitive nouns, such as public figures’ names, well-known place names, and well-known event names. In contrast, the scope of pronouns in argot recognition is broader. Therefore, compared to variant recognition tasks, both argot recognition and interpretation tasks become more complex and challenging. Based on these differences, Xu et al. [8] constructed a dataset of Chinese cant word-pronoun pairs for cant recognition tasks, providing evaluation support for future argot recognition tasks.

Compared with the field of Chinese argot recognition, there has been relatively more progress in the field of English argot recognition. Due to the dynamic and rapidly evolving nature of cybercrime, argot vocabulary undergoes continuous changes, with additions and deletions occurring. Additionally, each criminal group may establish its own industry-specific argot (e.g., drug traffickers) [17]. Consequently, there has been a shift towards machine learning methods for argot recognition, gradually replacing traditional manual construction of argot knowledge bases. In 2015, Dhuliawala et al. [18] proposed an English slang dictionary called SlangNet, aiming to complement WordNet for use in natural language processing (NLP) applications. The research team also evaluated the resource using the Lesk algorithm and the Extended Lesk algorithm. Furthermore, this work showed how to leverage online crowdsourcing resources to build high-quality language resources. In 2016, Wu et al. [19] constructed the slang dataset SlangSD for sentiment analysis of social media. Greg Durrett et al. [20] focused on the task of product keyword identification in online cybercrime forums and studied the effects of different research methods on product keyword identification through custom datasets. Later in 2018, Yuan et al. [5] proposed an argot recognition framework, Cantreader, incorporating improved word2vec and Hypernym identification, achieving commendable results in identifying English argot words across various forums on the dark web. In the 2020 study, Wilson et al. [21] used the Urban Dictionary dataset to train a set of word vectors and evaluated them in multiple slang-related tasks. The set of word vectors achieved significant improvements in specific tasks. Aravinda et al. [22], integrating language models and knowledge graphs, introduced a framework for detecting English slang in social media in 2022. Their approach demonstrated good performance in downstream experimental tasks, such as emotion detection, hate speech detection, and crime detection.

In summary, the current state of automated argot recognition faces several challenges:Lack of research and datasets specifically focused on argot recognition in the Chinese language domain.Existing studies on argot recognition are often domain-specific, lacking the development of a universally applicable framework for argot recognition across diverse domains.Most existing models rely on extensive prior data for training, hindering the generalization and cold start capabilities of argot recognition algorithms in unfamiliar domains.

Inspired by the manifold assumption [23], this paper posits that the commendable performance of numerous deep learning models based on word embeddings indicates that vectors obtained through word embeddings carry specific semantic meanings within their high-dimensional space. The lower the entropy of the set of vectors obtained through word embeddings, the more distinct the semantic meanings conveyed by the word embeddings, indicating a more effective performance of word embeddings.

Therefore, based on this assumption, to address the aforementioned issues, this paper has preliminarily established the Chinese argot dataset MNGG. Subsequently, the CSRMECT argot recognition model and the LLMResolve argot interpretation framework are proposed. Both the model and the framework are designed with a cold start approach, leveraging extensive knowledge embedded in pretrained texts to achieve generalization in unfamiliar domains, eliminating the need for domain-specific datasets for training.

## 3. Chinese Argot Recognition Based on CSRMECT Model

### 3.1. Entrophy Based Semantic Space

By combining the concept of information entropy from information theory with the semantic meanings of word vectors in word embeddings, this paper proposes a semantic space based on information entropy. This space is utilized to assess the semantic coherence and richness of word vectors.

#### 3.1.1. Vector Embedding and Semantic Space

In a series of papers around the year 2000, Joshua Bengio and others [24] employed neural probabilistic language models to enable machines to “learn a distributed representation for words”, thereby achieving the goal of dimensionality reduction in the word space. Subsequently, over the following decades, various well-known word-embedding algorithms emerged, including Word2Vec [25], GloVe [26], and others.

The essence of word embeddings lies in reducing words with rich semantics to vectors of specific dimensions, where each dimension carries a specific meaning, as shown in Figure 2.

#### 3.1.2. Word Vector Semantic Rationality Index Based on Information Entropy

In the task of argot recognition, each sentence is treated as a separate corpus for word embedding. Consequently, a single word may have multiple word vectors. This paper posits that when a vocabulary term is used as an argot, its spatial position in the semantic space should exhibit significant differences compared to its position when used in regular contexts alongside other non-argot words. Moreover, vectors generated by more advanced word-embedding algorithms should exhibit more pronounced spatial distribution characteristics, with vectors of words used in similar contexts converging together.

For example, the Chinese word “打击” has two meanings in English, namely “hit” and “catastrophe”. In an ideal word-embedding vector result, as shown on the left side of Figure 3, the vectors for these meanings should be distinct. By contrast, an undesirable result is depicted on the right side, where the vectors fail to adequately differentiate between the meanings.

It is evident that the higher the entropy of the set of word vectors corresponding to multiple usages of a word, the poorer the embedding performance of the current word-embedding algorithm for that specific usage of the word. To measure the entropy of a set of word vectors, we utilize the following formula based on the definition of information entropy:Retrieve the set of word vectors $S_{i}$ corresponding to the *i*-th usage from the word-embedding result $E_{word}$ of the word word.Let $C=\frac{\sum_{j=1}^{n}S_{i}}{n}$ denote the core vector.Calculate the distance $d_{i}=∥V_{i}-C∥$ for each vector $V_{i}$ in $S_{i}$ to the core vector.Assuming $P_{i}$ is the probability for the *i*-th vector, use the normalized exponential of the distance as the probability: $P_{i}=\frac{\exp(-\beta d_{i})}{\sum_{j}\exp\left(-\beta d_{j}\right)}$, where β is a parameter.Calculate the entropy of the vector set $S_{i}$ as $H_{i}=-\sum_{i}P_{i}\log\left(P_{i}\right)$.The vector information entropy for the word word in its word embedding is given by $Entropy\left(word\right)=\frac{\sum_{i=1}^{n}H_{i}}{n}$.

### 3.2. Enhanced MECT Model

The MECT model, proposed by Wu et al., is a cross-transformer based on multi-modal embeddings, applied in Chinese named entity recognition tasks [27]. As illustrated in Figure 4, the MECT model consists primarily of multi-modal embedding layers and cross-transformer layers. Previous studies have demonstrated the model’s commendable accuracy in identifying Chinese entities, efficient operational speed, and notable interpretability [28,29]. We will employ the MECT model for the word vector embedding tasks described above.

#### 3.2.1. Multivariate Data Embedding Layer

This layer comprises two main components: lattice embedding and Chinese radical-level embedding. Lattice embedding is a crucial element of the FLAT model [30], encompassing semantic and positional boundary information in the lattice data, comprehensively considering contextual features in sentences. Taking the sentence “Nanjing Yangtze River Bridge” as an example, the input situation for lattice embedding is illustrated in Figure 5, containing the head and tail positions of characters and words.

Chinese characters are based on ideograms, representing their meanings through the shapes of objects. For instance, characters with “艹” or “木” as radical components often represent plants and can effectively recognize raw materials used in the production of drugs, such as “cannabis” and “ephedra”. Characters with “月” as a radical component often represent body parts or organs and can adeptly identify euphemisms in the adult content domain. There are various methods for decomposing Chinese characters, including radical decomposition (CR), head-tail decomposition (HT), and structural decomposition (SC), as illustrated in Table 1.

To extract radical-level features of Chinese characters, an improved CNN network is constructed in this paper. CNN was initially proposed in the LeNet-5 model [31] and was applied in AlexNet in 2012 [32], achieving significant breakthroughs in the field of image recognition. Therefore, this paper selects the information-rich structural composition (SC) as the radical-level feature of Chinese characters and utilizes CNN to extract features of the characters. The specific process of embedding Chinese characters at the radical level is as follows:Decompose Chinese characters into radicals (SC) and input them into a CNN network.Embed radicals at the radical level for convolutional operations in the convolutional layer.Utilize max-pooling and fully connected layers to obtain the final embedding vector for Chinese character radicals.

#### 3.2.2. Cross-Transformer Layer

The MECT model introduces a cross-transformer network [27], as illustrated in Figure 6. This network employs two transformer encoders, which independently process information from lattice embeddings and Chinese radical embeddings. It achieves the enrichment of Chinese character semantic information by incorporating contextual and lexical information.

The inputs QL(QR), KL(KR), and VL(VR) in the cross-transformer network are obtained through linear transformations using lattice embeddings or Chinese radical embeddings, as defined in Equation (Equation 1).
(1)$${\left[\begin{array}{c} Q_{L(R),i}\\ K_{L(R),i}\\ V_{L(R),i} \end{array}\right]}^{T}=E_{L(R),i}\cdot {\left[\begin{array}{c} W_{L(R),Q}\\ E\\ W_{L(R),V} \end{array}\right]}^{T}$$

In the context, $E_{L,i}$ and $E_{R,i}$ represent the *i*-th lattice embedding vector and Chinese radical embedding vector, respectively. Here, *E* denotes the unit vector, and *W* represents learnable parameters. In the cross-transformer network, the attention calculation formula is given by:(2)$$Att\left(A_{R},V_{L}\right)=Softmax\left(A_{R}\right)V_{L}$$
(3)$$Att\left(A_{L},V_{R}\right)=Softmax\left(A_{L}\right)V_{R}$$
(4)$$A_{L\left(R\right),ij}=\epsilon \left(u\right)K_{R\left(L\right),j}+\epsilon \left(v\right){R^{*}}_{L\left(R\right),ij}$$

Wherein, the lower-left corner’s *L* denotes the values from the lattice embedding side, and *R* represents the values from the Chinese radical embedding side. The parameters *u* and *v* in Formula (4) represent learnable attention offset parameters. Here, $R_{ij}^{*}=R_{ij}\cdot W$, where *W* is a learnable parameter, and $\epsilon \left(x\right)={\left(Q_{L(R),i}+x_{L(R)}\right)}^{T}$. The calculation of $R_{ij}$ is as follows:(5)$$R_{ij}=ReLU\left(W_{r}\left(p_{h_{i}-h_{j}}⨁p_{t_{i}-t_{j}}\right)\right)$$
(6)$$p_{span}^{\left(2k\right)}=\sin\left(\frac{span}{10000^{\frac{2k}{d_{model}}}}\right)$$
(7)$$p_{span}^{\left(2k+1\right)}=cos\left(\frac{span}{10000^{\frac{2k}{d_{model}}}}\right)$$

Among these, $R_{ij}$ represents the computation of the relative distance between positions *i* and *j*, where $W_{r}$ denotes a learnable parameter, and $h_{i}$ and $t_{i}$, respectively, signify the head and tail positions of the Chinese character at position i. The symbol ⨁ signifies a concatenation operation. In Formulas (6) and (7), the term span corresponds to $h_{i}-h_{j}$ or $t_{i}-t_{j}$ as defined in [33].

#### 3.2.3. CSRMECT Model

To integrate word vectors considering both the context and Chinese character structure, this paper proposes the CSRMECT model, building upon the MECT model with modifications. Specifically, we enhance the MECT model by removing the final CRF layer and directing the output character vectors from the linear layer into the vector aggregator module for word vector synthesis. The final output is a context-encoded word vector, as illustrated in Figure 7. The vector aggregator module takes character vectors as input and produces word vectors. In this paper, we adopt the default approach for constructing the vector aggregator module, as depicted in the pseudocode in Algorithm 1.
**Algorithm 1:** Vector Aggregator
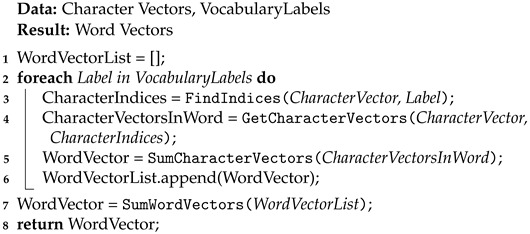


The field of argot recognition has long been plagued by the lack of high-quality annotated datasets. Through the above changes, the CSRMECT model successfully solved this problem. Specifically, the training of the CSRMECT model only requires the use of a normal corpus. The trained CSRMECT model can understand the contextual relationships in the sentence and output a word vector with contextual semantics for each word.

### 3.3. DBSCAN Clustering Algorithm

Clustering, one of the primary methods for knowledge discovery in large datasets, encompasses various prevalent techniques in the field of semantic clustering, including k-means [34], hierarchical clustering [35], and DBSCAN [36]. The k-means algorithm exhibits limitations in handling non-spherical clusters and is susceptible to the choice of initial cluster centers [37], necessitating the pre-specification of the cluster quantity (K value). Hierarchical clustering, often proceeding in a top-down or bottom-up hierarchical decomposition due to its simplicity, tends to form cluster chains. In contrast, DBSCAN possesses the capability to cluster shapes of arbitrary forms, such as linear, concave, elliptical, without the need for predefined cluster quantities. Additionally, DBSCAN has been proven effective in handling massive databases [36,38,39]. Consequently, we employ the DBSCAN algorithm for clustering, aiming to extract argot vocabulary from extensive sets of word vectors.

The DBSCAN clustering algorithm determines the density of a point by calculating the number of points within a specified radius. Points with densities exceeding a specified threshold are grouped into clusters. Given the high dimensionality and sparsity between word vectors in this research, the Euclidean distance proves inadequate for accurately measuring the vector similarity. Hence, the cosine distance is chosen for DBSCAN clustering, with the formula as follows:(8)$$D\left(p,q\right)=\frac{\sum_{i=1}^{n}\left(p_{i}\times q_{i}\right)}{\sqrt{\sum_{i=1}^{n}{\left(p_{i}\right)}^{2}\times \sum_{i=1}^{n}{\left(q_{i}\right)}^{2}}}$$

In the context where *q* and *p* represent arbitrary word vectors, and *n* denotes the dimensionality of the word vectors, with $p_{i}$ and $q_{i}$ indicating the values in the *i*-th dimension of the word vectors, the workflow of the DBSCAN clustering algorithm is outlined as follows:Randomly select a word vector *q* as the object, defining its neighborhood as $E_{q}$, and compute the cosine distance values between it and other word vectors *p*.If $D\left(p,q\right)<\varepsilon$, categorize the word vector *p* into $E_{q}$. If $D\left(p,q\right)>\varepsilon$, ignore the word vector *p*.Tally the number of word vectors in $E_{q}$. If $count\left(E_{q}\right)>minpts$, designate $E_{q}$ as a cluster and recursively process other word vectors in the same manner. Otherwise, label the word vector as noise data.

Here, ε represents the scanning radius distance, and minpts stands for the minimum number of enclosed points. Both are selectable parameters.

### 3.4. Chinese Argot Recognition Work

The CSRMECT model, tailored to the structural similarities between Chinese argot vocabulary and reference terms, as well as the contextual disparities with the original meanings of argot vocabulary, involves a two-stage process. This process incorporates an enhanced MECT model for the fusion of contextual and Chinese character structural features in word vector representation and utilizes the DBSCAN clustering algorithm for the discovery of semantically inconsistent argot vocabulary in the semantic space. The specific workflow is illustrated in Figure 8.

Firstly, normal corpora and argot corpora (dataset containing argots) are amalgamated into datasets. The CSRMECT model extracts lattice embedding vectors and Chinese radical-level embedding vectors from sentences, followed by a fusion operation. Subsequently, through context encoding, word vector representations for each Chinese vocabulary in the sentence are obtained.All word vectors derived from the processed normal corpus dataset *N* are mapped to a high-dimensional space. The DBSCAN clustering algorithm is employed to partition various clusters, yielding the core cluster vector set N(W) for each vocabulary *W* in dataset *N*.For the argot corpus dataset *M*, all word vectors are similarly mapped to a high-dimensional space, resulting in the high-dimensional word vector M(W) for each vocabulary *W* in dataset *M*.In the vocabulary list $M_{List}$ of argot corpus *M*, the label list is computed as $L_{M}=\left[\mathrm{Label}\left(W_{i}\right)|W_{i}\in M_{List}\right]$, where $\mathrm{Label}\left(W_{i}\right)$ is determined as follows:
$$\mathrm{Label}\left(W_{i}\right)=\left\{\begin{array}{cc} 1, & \mathrm{if}\;\forall q\in N\left(W_{i}\right):D\left(M\left(W_{i}\right),q\right)>\varepsilon \\ 0, & \mathrm{if}\;\exists q\in N\left(W_{i}\right):D\left(M\left(W_{i}\right),q\right)\leq \varepsilon \end{array}\right.$$At this time, in the list $L_{M}$, the words marked 1 are argots, and vice versa for normal words.

It is worth mentioning that in this code word recognition framework, the MECT model in the above process exists as a natural language deep learning model. This means that when a better model appears in the future, that model will be able to replace the MECT model here, allowing the clustering algorithm to obtain better word vectors. We believe that the rapid development in the field of deep learning will directly promote the progress of the field of argot recognition through this framework.

## 4. Argot Interpretation Based on Large Language Models

### 4.1. Large Language Models

Since the emergence of ChatGPT in 2022 [40], a plethora of research has surfaced regarding the integration of large language models with various classical machine learning tasks to enhance their effectiveness. Yang et al. introduced the PICa few-shot prompting method, applying large language models as knowledge bases in the VQA domain [41]. Building upon PICa, Shao et al. proposed the Prophet few-shot prompting method, achieving commendable performance in VQA by leveraging answer heuristics to prompt GPT-3 [42]. OpenAI’s experiments indicate that simply scaling up language models significantly improves their performance in NLP tasks, such as knowledge-based QA and language understanding [40]. Chen et al. constructed the Codex programming assistance tool based on large language models and prompt engineering, addressing 70.2% of programming problems in testing [43]. Sun et al. tested ChatGPT’s retrieval capability with successful outcomes [44].

Through resource-intensive training, large language models embed a substantial amount of prior knowledge from the corpus into their parameters. Consequently, large language models can function as knowledge engines, providing external knowledge to enhance task performance across various machine learning tasks. For specific domains, fine-tuning the model with domain-specific texts significantly enhances its understanding of that domain, thereby improving task performance.

### 4.2. Argot Interpretation Based on Large Language Models

Building on the aforementioned analysis, this paper employs large language models for argot interpretation. Specifically, to investigate the feasibility of using large language models for argot interpretation, this study leaves argot vocabulary blank in MNGG. Through the prompt engineering, syntactic information and cue words are conveyed to the large language model. The vast prior knowledge acquired during the pretraining of the large language model is utilized for the task of argot interpretation.

## 5. Experimental Process

### 5.1. Datasets and Parameters

#### 5.1.1. MNGG Argot Recognition Evaluation Dataset

Building upon the achievements of the dogwhistle dataset in the work by Xu et al. [8], this paper integrates argot corpora to create the MNGG (Mystique Naming Glossary Gathering) dataset. The task of argot recognition is transformed into a sequence labeling task for training and testing. From the Insider and Outsider subtasks of the dogwhistle dataset, the paper extracts pairs of argots and referential terms, resulting in a total of 1684 annotated argot-referential term pairs. Leveraging these argot pairs, the paper utilizes Chinese text data from THUC News [45] as the base text and replaces referential terms in the base text with argot words from the pairs. After this replacement operation is completed, we use the BIO sequence annotation method to mark the argots in the sequence. As shown in Figure 9, this process produces an annotated argot corpus with contextual information.

To facilitate the verification of the training data’s impact on model performance and to enable rapid testing with reduced data, the MNGG dataset also includes a clipped subset. This subset, denoted by the .clip file name suffix, represents a 10% extraction from the complete dataset. The number of corpora in the MNGG dataset is presented in Table 2.

#### 5.1.2. Enhanced Base Corpora Utilizing Wikipedia and Large Language Models

In order to obtain clustering results for normal text, facilitating subsequent labeling of the argot dataset using clustering algorithms, this paper leverages Wikipedia text to establish a base corpus. Additionally, the paper employs prompt engineering to enhance the base corpus with a massive text dataset, effectively addressing the presence of vocabulary beyond the base corpus in the argot dataset. Combined with the CSRMECT model, this paper generates a substantial collection of high-dimensional word vectors based on words present in the normal corpus of the base dataset. An overview of the base corpora is provided in Table 3.

Here, the inclusion rate in argot denotes the ratio of the number of words in the argot corpus that appear in the base corpus to the total number of words in the argot corpus. By utilizing large language models and iteratively invoking prompts as illustrated in Figure 10, this paper cleans and organizes the obtained data to construct the LLM enhancement corpus, enhancing the inclusion rate of argot vocabulary and the occurrence frequency of argot terms.

### 5.2. Argot Recognition Experiment

#### Metric Calculation Based on BIO-Format Sequence Labeling

The BIO annotation scheme is a labeling method introduced and utilized in the field of named entity recognition (NER), indicating whether words or tokens in the labeled sequence belong to an entity. It has become a common annotation scheme in the field of natural language processing. Its design aims to distinguish the beginning (B: Beginning), interior (I: Inside), and non-entity (O: Outside) parts of an entity.

Calculating metrics for BIO-annotated sequences involves comparing the similarity between predicted and actual sequences. Specifically, given the predicted and actual sequences, the ranges of labeled entities in both sequences can be statistically determined. Subsequently, Precision, Recall, and the F1 are computed to evaluate the effectiveness of the predicted sequence. These metrics are calculated using the following formulas:(9)$$Precision=\frac{TP}{TP+FP}$$
(10)$$Recall=\frac{TP}{TP+FN}$$
(11)$$F1=2\frac{Precision\times Recall}{Precision+Recall}$$

Here, TP represents the number of correctly predicted positive instances, FP denotes the number of incorrectly predicted positive instances, and FN stands for the number of positive instances that were not predicted.

### 5.3. Experimental Results

In the work by Zhang et al. [14], the SVM classifier, enriched with additional feature extraction from text, achieved notable recognition rates in the field of argot recognition. This study replicates the SVM classifier mentioned in the literature, as the classifier is applied and tested on the MNGG dataset. MNGG transforms the argot recognition experiment into a sequence labeling algorithm. Consequently, this paper explores various classical sequence labeling algorithms, annotates the MNGG dataset using the BIO format, and tests the algorithmic performance. The comparative effectiveness of these algorithms with the proposed CSRMECT model is presented in Table 4.

### 5.4. Argot Interpretation Experiment

To investigate the feasibility of using large language models as knowledge engines for argot interpretation, this study conducted argot interpretation experiments based on the MNGG dataset, which contains a total of 1684 pairs of argots. GPT-3, GPT-4, and prompt engineering were employed in the experiments.

The specific experimental procedure is as follows:Split all texts in the MNGG dataset into sentences. Extract sentences containing only one argot from the split corpus, denoted as the corpus *W*.For each sentence $W_{i}$ in the corpus, extract the argot $word_{i}$ from it. Randomly select T−1 words from the argot vocabulary of the MNGG dataset, forming a list of prompt words $Lst_{i}$.Utilizing the prompt engineering approach shown in Figure 11, input both syntactic information and the prompt word list into the large language model, obtaining the judgment corpus $P_{i}$.Tokenize $P_{i}$ to obtain the tokenized vocabulary set $\bar{P_{i}}$.For similarity measurement, train a word2vec model using the Wiki + LLM base corpus. Let the word vector for the vocabulary *X* in the word2vec model be $\vec{X}=word2vec\left(X\right)$.For the *i*-th sentence $W_{i}$ and its judgment corpus $\bar{P_{i}}$, if there exists a vocabulary $p_{i,j}\in \bar{P_{i}}$ satisfying $eps\leq \frac{word2vec\left(p_{i,j}\right)\cdot word2vec\left(word_{i}\right)}{||word2vec\left(p_{i,j}\right)\left|\right|\cdot ||word2vec\left(word_{i}\right)\left|\right|}$, the *i*-th argot recognition is considered successful; otherwise, it fails.For all sentences $W_{i}$ in the corpus *W*, calculate its accuracy $acc=\frac{count(W_{success})}{count(W)}$.

The experimental results are presented in Table 5.

## 6. Analysis and Discussion

### 6.1. Discussion on Argot Recognition Experiments

The effectiveness of the CSRMECT model is contingent upon prior conditions, such as the size of the base corpora and the model parameter settings. This section provides a discussion of such issues.

#### 6.1.1. Qualitative Analysis of Clustering Results

The CSRMECT model is employed in this study to obtain word vectors. To analyze the semantic richness of word vectors, the t-SNE dimensionality reduction algorithm [49] is employed. Quantitative analysis is performed on selected argot and non-argot words from certain base datasets. The results are illustrated in Figure 12.

In Figure 12, the term “客场” (can be translated into opponent’s field, away, etc.) is employed as an argot in every sentence it appears, whereas the term “自己” (can be translated into self, oneself, etc.) is a regular word. In the clustering results on the left side of the figure, the distance between the red points representing argots and the blue points representing normal words is about 10 to 100. In contrast, the distance between the red point and the blue point in the picture on the right is about 0.5 to 1. It is evident that there exists semantic differentiation in the spatial representation between argot and non-argot words. Through DBSCAN clustering, we can easily identify the vast majority of argots on the left side.

#### 6.1.2. Analysis of Data Augmentation Effects

As the semantic nature of vocabulary reflects in the relative positioning within the word vector space, the usage of argot terms in the base corpus directly impacts the effectiveness of argot recognition. When there is a scarcity of argot terms or their usage is overly uniform, the overall algorithmic process may be compromised. This study employs an augmentation algorithm based on large language models, enhancing the diversity of argot term usage in the base corpus to improve evaluation outcomes. The cumulative information on successive base data augmentation and corresponding algorithmic improvements is presented in Table 6.

#### 6.1.3. Sensitivity Analysis

The DBSCAN clustering algorithm involves two hyperparameters: minpts and ε. Here, minpts represents the minimum number of points within the same cluster, and ε indicates the size of the clustering boundary. To explore the model’s sensitivity to hyperparameters, including ε, and ensure experimental efficiency, word vectors need to be dimensionally reduced and then subjected to the DBSCAN clustering algorithm. Common dimensionality reduction algorithms include PCA [50], t-SNE, and others. In this study, a subset of vocabulary word vectors is selected, and different algorithms are employed to reduce the vectors to two dimensions for visualization, as depicted in Figure 13.

In the figure, the number of clusters before dimensionality reduction for the *i*-th row data is *i*. It is observed that the PCA, MDS [51], and lsomap algorithms show similar effects, while the t-SNE and UMAP [52] algorithms exhibit comparable effects and better performance.

Based on the above analysis, the PCA and t-SNE algorithms are selected for experimentation to investigate the model’s sensitivity to hyperparameters ε, dimensionality reduction algorithms, and reduced dimensions, as shown in Figure 14.

The experimental results indicate that around the optimal ε value, the stability of the F1-score is high. As ε deviates from the optimum, the F1-score gradually decreases. Additionally, due to the adoption of dimensionality reduction algorithms, while the model’s operational efficiency significantly improves, there is a slight reduction in model accuracy. Furthermore, compared to data augmentation methods, the choice of dimensionality reduction algorithms and reduced dimensions has a relatively minor impact on the F1-score.

For the vector aggregator module in CSRMECT, this study explores various implementation approaches and conducts sensitivity tests, as illustrated in Figure 15.

Here, “Sum” refers to adding multiple word vectors of a single word to form the word vector, while “Average” refers to summing and averaging multiple word vectors of a single word to obtain the word vector. In comparison to data augmentation methods, the modification of the vector aggregator implementation has a minimal impact on the results. However, it is notable that the model’s performance improves consistently across different parameters when using the Average algorithm. Hence, it cannot be ruled out that a more rational vector aggregator implementation could significantly enhance the model’s effectiveness.

Another noteworthy observation is that when calculating the rationality of word vectors under different vector aggregators using the information entropy mentioned earlier, the set scores obtained with the “Sum” method tend to be higher than those with the “Average” method, as shown in Table 7. This suggests that more rational word vectors indeed contribute to the improvement of model performance.

### 6.2. Discussion on Argot Vocabulary Interpretation Experiment

To analyze the recognition effectiveness of the LLMResolve framework, and to explore the strengths and limitations of large language models in the field of argot recognition, statistical and qualitative analyses are employed to discuss the experimental results.

#### Statistical Analysis

Different large language models exhibit varying performance; theoretically, utilizing more advanced models enhances the task of argot interpretation. A comparison is made between GPT-3.5 and GPT-4, as shown in Table 8.

The results indicate superior performance using GPT-4 compared to GPT-3.5. Thus, in LLMResolve, the performance of large language models significantly influences the accuracy of argot resolution. At the same time, we also found that for large language models, the performance of LLMResolve is not good when there is no hint word. However, from a certain perspective, this is also normal because even for humans, achieving this is very difficult.

POS-tagging is conducted using the jieba tool, with codes corresponding to the meanings detailed in Table 9.

Combining the output results of the LLMResolve framework, a statistical analysis is conducted on the common POS tags, yielding recognition quantities and rates for each POS, as illustrated in Figure 16.

The MNGG dataset proposed in this paper, in contrast to the Weibo dataset used in previous studies [12], differs in that the argot vocabulary in MNGG extends beyond a small subset of nouns like personal and location names. It encompasses various parts of speech, including nouns, verbs, adjectives, and adverbs. Analyzing the figure above reveals that the diversity of parts of speech poses certain challenges for argot interpretation work, with occasional recognition failures observed for verbs and adverbs. Simultaneously, as the most frequently occurring nouns (i.e., labeled as n, nr, ns), their recognition rate averages 82.4%, comparatively lower within the spectrum of parts of speech recognition rates.

### 6.3. Qualitative Analysis

In conjunction with the foregoing, this paper conducts a qualitative analysis on failed instances of noun disambiguation. Failed samples extracted from the experimental results are presented in Figure 17.

Summarizing from the table, the following reasons for disambiguation failure can be delineated:**Insufficient Contextual Information**: In the case of Sample 1, the phrase “蠢蠢的钟摆” (an animated pendulum) employs personification in Chinese rhetoric. Without ample contextual cues, both large language models and humans struggle to accurately discern the intended word for this context. For Sample 2, where both “天气” (weather) and “市场” (market) share the characteristic of change, additional context is essential for auxiliary reasoning.**High Similarity of Prompt Words**: Illustrated by Sample 3, the words “鲜红色” (crimson) and “白色” (white) both represent colors, making it challenging for the model to distinguish their semantic differences within the sentence.**Triggering Safety Mechanism in Large Language Models**: When sensitive terms appear in the prompt engineering, the safety mechanism of large language models is triggered. Consequently, the model refrains from providing an effective response and instead elaborates on the reason for refusing to answer. This phenomenon is particularly prevalent in the context of drug-related or explicit argots.

## 7. Conclusions

This study introduces, for the first time, the concept of utilizing semantic conflicts in argot vocabulary for argot recognition. Leveraging the MECT model, we propose the CSRMECT model for argot recognition and employ LLMResolve for argot interpretation. The proposed argot recognition and interpretation models surpass previous research efforts. Extensive experiments in this study provide insightful analyses of the model performance.

In terms of argot recognition, experiments indicate that improving the rationality of word vectorization methods enhances argot recognition. Furthermore, under the same vectorization algorithm, the similarity between argots and surrounding sentences also influences argot recognition effectiveness. Regarding argot interpretation, the outstanding performance of large language models validates their feasibility as knowledge engines for argot interpretation. Additionally, experiments demonstrate that more powerful models offer stronger background knowledge and better argot recognition capabilities.

For the future development of argot recognition models, a primary task is to investigate more rational word vectorization algorithms to expand the semantic space gap between argot and general vocabulary, thereby improving recognition rates. As for argot interpretation tasks, feasible future research directions include fine-tuning large language models using known argot repositories and exploring methods to bypass security mechanisms in large language models to enhance model response rates.

## Figures and Tables

**Figure 1 entropy-26-00321-f001:**
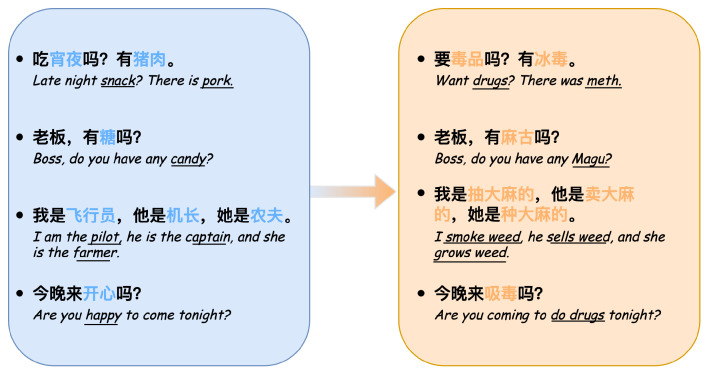
Examples of usage of argots.

**Figure 2 entropy-26-00321-f002:**
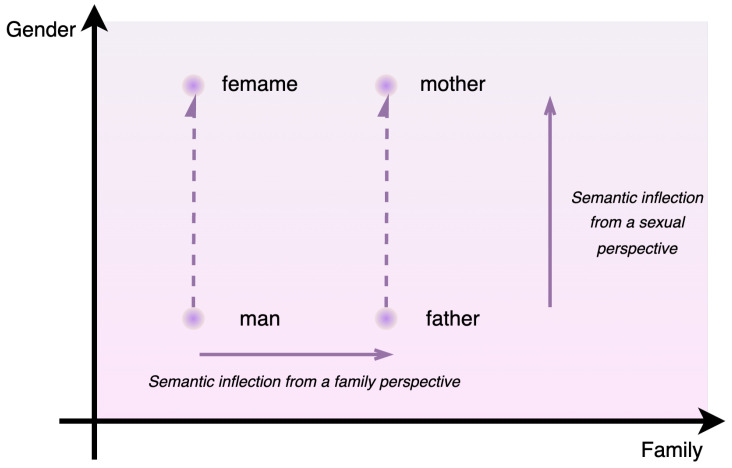
Word embedding example.

**Figure 3 entropy-26-00321-f003:**
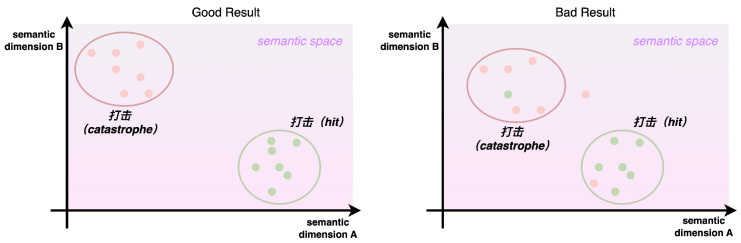
Comparison of word-embedding effects.

**Figure 4 entropy-26-00321-f004:**
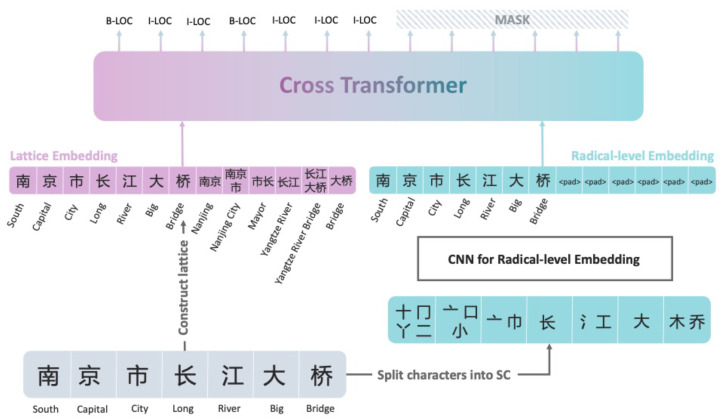
MECT workflow.

**Figure 5 entropy-26-00321-f005:**
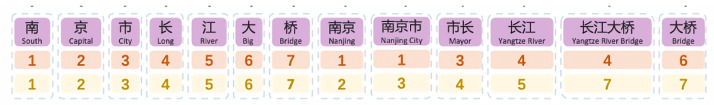
MECT Lattice Embedding.

**Figure 6 entropy-26-00321-f006:**
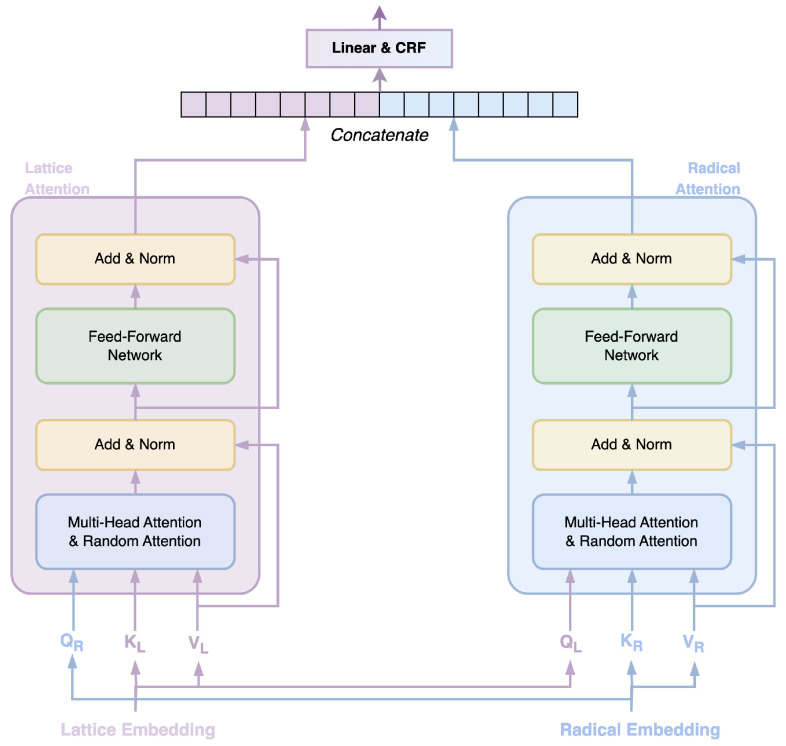
Cross-transformer layer.

**Figure 7 entropy-26-00321-f007:**
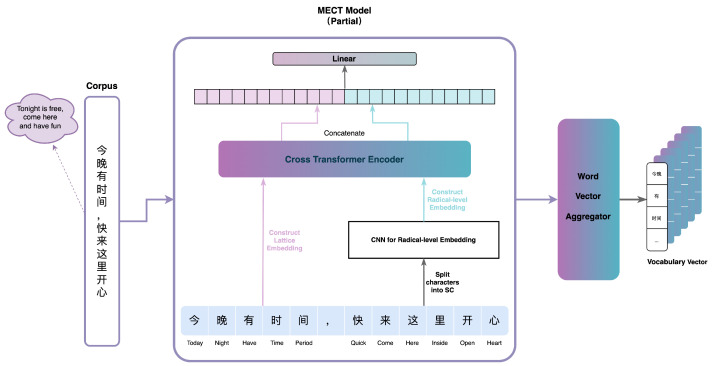
CSRMECT model.

**Figure 8 entropy-26-00321-f008:**
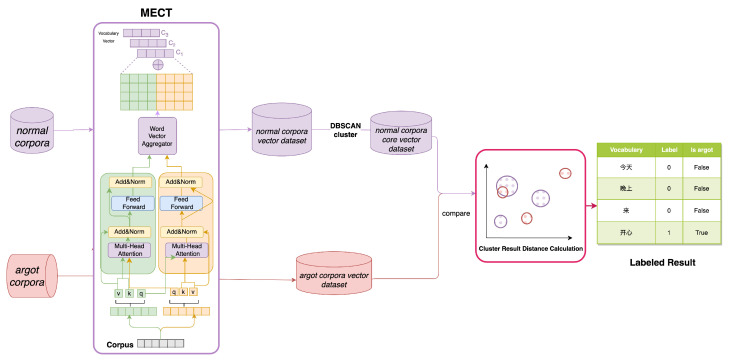
Specific workflow of argot discovery.

**Figure 9 entropy-26-00321-f009:**
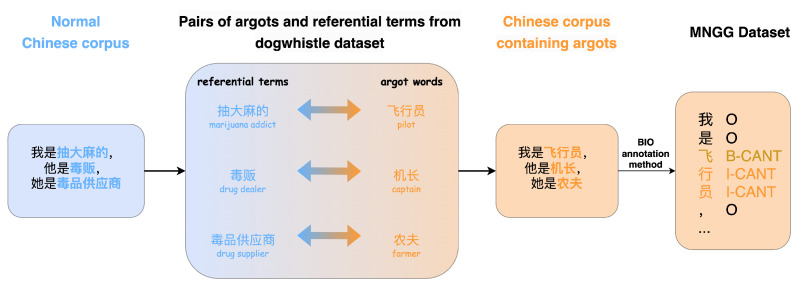
Build MNGG dataset.

**Figure 10 entropy-26-00321-f010:**
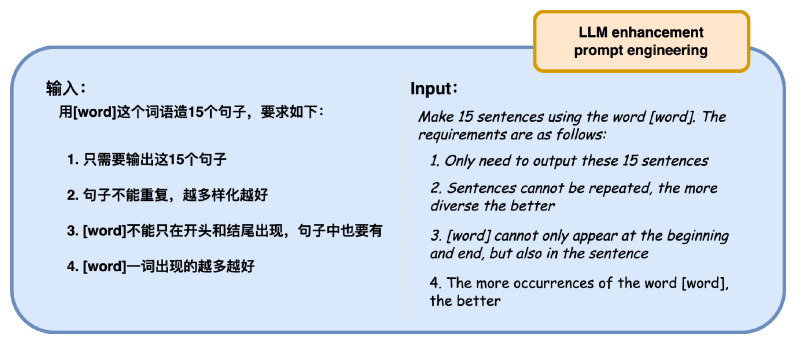
Prompt for LLM enhancement corpus.

**Figure 11 entropy-26-00321-f011:**
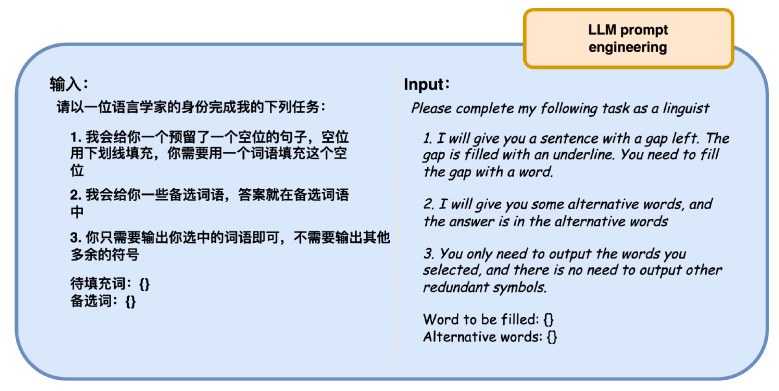
Inputting syntactic information and prompt word list into the large language model.

**Figure 12 entropy-26-00321-f012:**
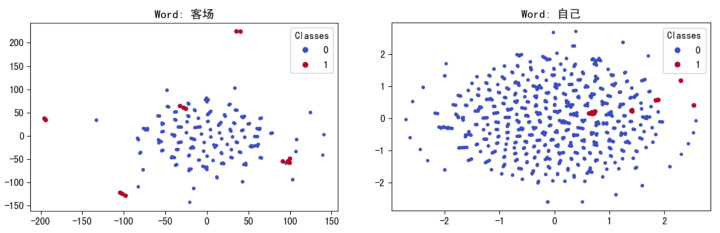
Dimensionality reduction visualization.

**Figure 13 entropy-26-00321-f013:**
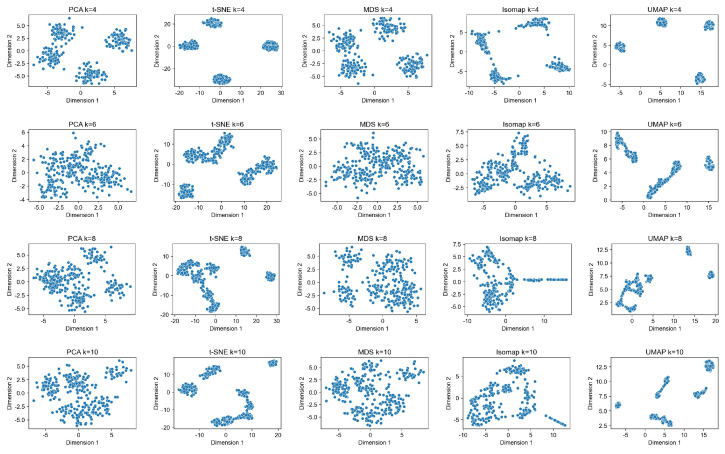
Comparison of dimensionality reduction algorithms.

**Figure 14 entropy-26-00321-f014:**
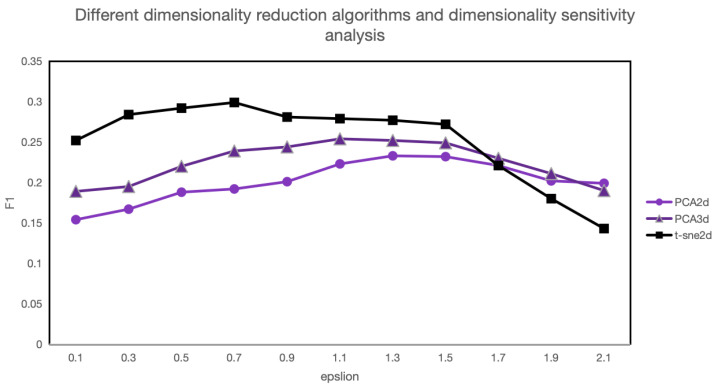
Sensitivity analysis for ε.

**Figure 15 entropy-26-00321-f015:**
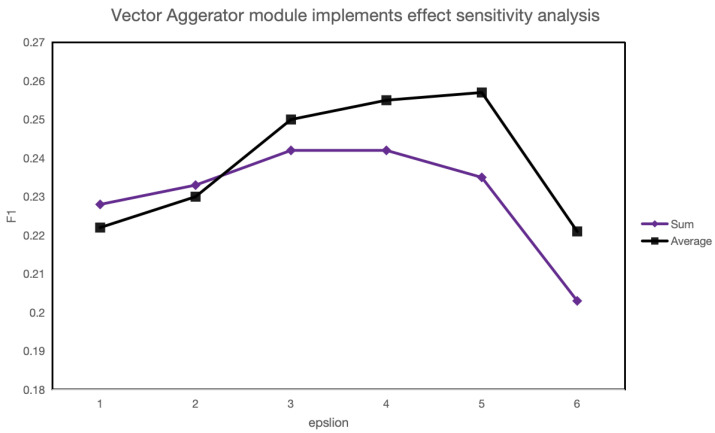
Sensitivity analysis of vector aggregator.

**Figure 16 entropy-26-00321-f016:**
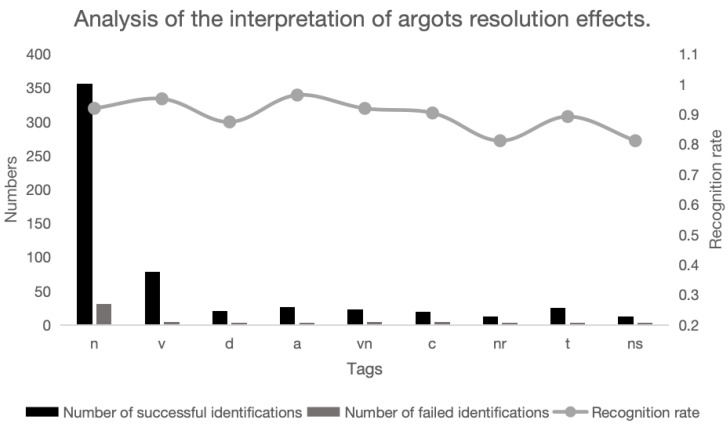
Argot interpretation effect POS analysis.

**Figure 17 entropy-26-00321-f017:**
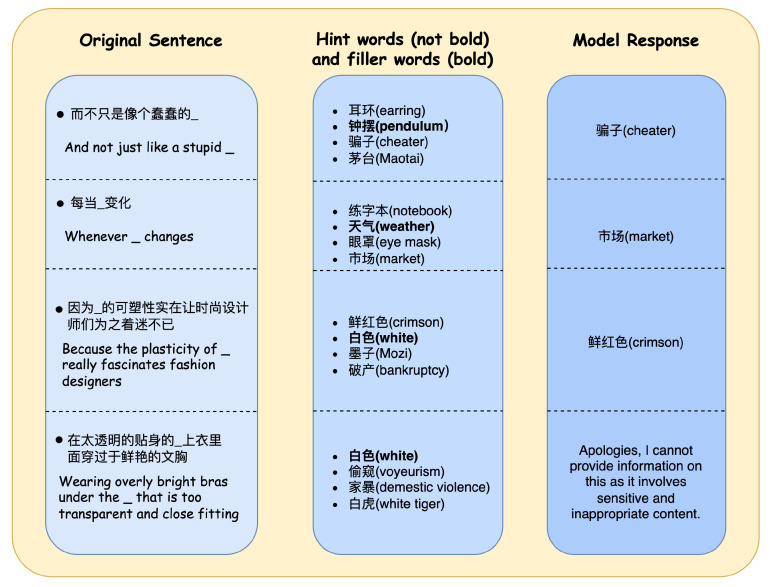
Failed examples of argot interpretation in LLMResolve.

**Table 1 entropy-26-00321-t001:** Character structure decomposition table.

Chinese Character	CR	HT	SC
麻 (numb)	广 (wide)	广林 (forest)	广木木 (wood)
蠕 (worms)	虫 (insect)	虫需 (need)	虫雨 (rain) 而 (but)
挂 (hang)	扌 (hand)	扌圭 (Gui)	扌土 (earth) 土
唱 (sing)	口 (mouth)	口昌 (thriving)	口曰 (speak) 曰

**Table 2 entropy-26-00321-t002:** Overview of the MNGG dataset.

Dataset	Number of Sentences	Number of Argot Vocabulary	Average Argot Vocabulary per Sentence
train.clip.bio	564	1116	1.97
test.clip.bio	338	705	2.09
dev.clip.bio	225	470	2.08
train.bio	5645	11,595	2.05
test.bio	3387	7315	2.15
dev.bio	2258	4751	2.10

**Table 3 entropy-26-00321-t003:** Overview of base corpora.

Base Corpus	Number of Entries (Sentences)	Inclusion Rate in Argot	Average Occurrence Frequency of Argot Vocabulary
Wiki	194,749	0.71	103.4
LLM	159,277	0.45	3619.9
Wiki + LLM	354,026	0.99	1685.5

**Table 4 entropy-26-00321-t004:** Comparison of argot recognition models.

Model	F1	Precision	Recall
SVM [14]	0.08	0.08	0.08
LGN [46]	0.03	0.78	0.02
Lattice-LSTM [47]	0.15	0.62	0.08
LR-CNN [48]	0.23	0.64	0.14
CSRMECT	0.33	0.35	0.31

**Table 5 entropy-26-00321-t005:** Argot interpretation experiment results.

Model	Accuracy
Huang2013 [12]	0.364
Zhang2015 [14]	0.383
Sha2017(Acc@20) [15]	0.870
LLMResolve	0.919
LLMResolve (GPT-4+10 Prompt Words)	0.824
LLMResolve (GPT-4+3 Prompt Words)	0.919

**Table 6 entropy-26-00321-t006:** Comparison of base data augmentation information and effects.

Base Corpus	Argot Loss	F1	P	R
Wiki	3145	0.04	0.22	0.02
LLM *	1918	0.27	0.57	0.18
LLM **	408	0.32	0.36	0.28
LLM ***	152	0.33	0.35	0.31

LLM * denotes augmentation once, LLM ** denotes augmentation twice, and LLM *** denotes augmentation three times.

**Table 7 entropy-26-00321-t007:** Evaluation results of using vector information entropy for different vector aggregators.

Word	Average	Sum
我们 (we)	2.943	9.643
我 (I)	2.707	10.223
一种 (a kind of)	2.887	8.224
方法 (method)	3.040	6.640
Average entropy value in dataset	2.902	3.455

**Table 8 entropy-26-00321-t008:** Experimental results of code interpretation under different large language models.

Model	Top-k	Accuracy
GPT-4	3	0.919
GPT-4	10	0.824
SOTA	-	0.919
GPT-3.5	3	0.768
GPT-3.5	5	0.741
GPT-3.5	10	0.648
GPT-3.5	20	0.537
GPT-3.5	30	0.454
GPT-3.5	0	0.133
SOTA	-	0.768

**Table 9 entropy-26-00321-t009:** Chinese POS tags and meanings.

POS Tag	Meaning	Detailed Meaning
n	Noun	Represents people, things, places, etc.
v	Verb	Indicates action, state, or behavioral existence.
d	Adverb	Used to modify verbs, adjectives, other adverbs, etc.
a	Adjective	Describes the qualities or states of things.
vn	Noun-Verb	Sometimes represents a mixture of nouns and verbs, typically used as a noun.
c	Conjunction	Connects words, such as “and”, “or”, etc.
nr	Name	Represents personal names.
t	Time Word	Represents words related to time.
ns	Place Name	Represents names of places.

## Data Availability

The research data supporting the reported results in this article are available on GitHub at the following link: https://github.com/Andrew82106/MNGG_Dataset (accessed on 24 November 2023).

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
