# Peer review of "Research on a Framework for Chinese Argot Recognition and Interpretation by Integrating Improved MECT Models"

_entropy, 2024, doi:10.3390/e26040321_

Round 1

Reviewer 1 Report

Comments and Suggestions for Authors

This paper presents a frame to identify and interpret argots in underground industries. Its main contributions are 1) constructing the MNGG dataset for the Chinese argot identification and interpretation 2) proposing the CSRMECT model for argot recognition and the LLMResolve model for argot interpretation. The identification and interpretation models outperform the existing models. The strategized vector construction method helps improve the argot identification performance, and the application of LLMs help improve the argot interpretation performance. The authors provide a detailed analysis of the model's performance  and a qualitative examination of failed instances in argot interpretation.

It's an interesting and important task, and the work itself showcased its effectiveness. It would be interesting to have more introduction about the current work along this line in non-Chinese languages. Also, the introduction to the cross-transformer (fig 3 specifically) was borrowed from another work which should have been cited properly or described in your own way. 

  Comments on the Quality of English Language

The English language needs to be improved. The punctuations are not used properly (e.g., you should have left a space between sentences). The way of citations should be re-edited. 

Reviewer 2 Report

Comments and Suggestions for Authors

In this paper, the authors presented a new Chinese argot dataset called MNGG, a model for Chinese argot recognition called CSRMECT and an argot interpretation framework called LLMResolve. LLMResolve leverages large language model and Prompt engineering for argot interpretation task. Experimental results verify the effectiveness of CSRMECT and LLMResolve. The topic is interesting and the detailed comments and some suggestions are listed below:

1. This paper proposes CSRMECT model for argot recognition. CSRMECT is a combination of MECT model and DBSCAN clustering algorithm, the authors only remove the origin CRF layer of MECT model and use a Vector Aggregator module instead, thus the innovation is relatively limited.

2. The authors should introduce the specific details of MNCG dataset. How the argots are annotated in the dataset should be illustrated, i.e. the label strategy should be introduced. The authors are encouraged to showcase the samples of the annotated data, like that of the NER annotation.

3. The paper has some variables without explanation.

Line 313, what does q refer to in this Equation?

Table 3, why there is a .clip version of the dataset?

4. Figure 10, Figure 11, Figure 12 are too blurry to recognize the text within. The authors should explain the differentiation in Figure 11 in a more detailed way.

Comments on the Quality of English Language

There are too many syntax errors and typos, the authors should checked manuscript thoroughly, e.g. 

(1) Line 123, extra 。

(2) The   in Fig.3 and 4 should be Nan and Jing.

(3) Today night have ... is weird in Fig.6. The exact translation can be listed in the Corpus frame.  

(4) The sentences and words in Chinese in Table 1,8,11 and Fig. 8,9,11 should also showcase the translation in English.

Round 2

Reviewer 2 Report

Comments and Suggestions for Authors

 I think the authors have addressed the issues proposed in the previous comments. I recommend the manuscript under consideration of acceptance after minor revision.

1.  The "Today Night ..." should be removed from Fig.7 unless the authors could provide more accurate translation of the Chinese sentence. The current translation will confuse the interested readers. Please try to make the bilingual sentences align in both semantics and the size.  

2. In Table 5, the accuracy of GPT-4+10 Prompt words is greatly lower than GPT-4+3 Prompt words. It seems counterintuitive. More analysis and discussions should be provided. 

Comments on the Quality of English Language

Moderate editing of English language required.